# Liver Transplantation for the Cure of Neuroendocrine Liver Metastasis: A Systematic Review with Particular Attention to the Risk Factors of Death and Recurrence

**DOI:** 10.3390/biomedicines12112419

**Published:** 2024-10-22

**Authors:** Quirino Lai, Alessandro Coppola, Anna Mrzljak, Maja Cigrovski Berkovic

**Affiliations:** 1General Surgery and Organ Transplantation Unit, Department of General and Specialty Surgery, Sapienza University of Rome, 00161 Rome, Italy; lai.quirino@libero.it; 2Department of General Surgery, Sapienza University of Rome, 00161 Rome, Italy; aless.coppola@uniforma1.it; 3Department of Gastroenterology and Hepatology, Liver Transplant Center, University Hospital Centre Zagreb, 10000 Zagreb, Croatia; anna.mrzljak@gmail.com; 4University of Zagreb, Department of Medicine, School of Medicine, 10000 Zagreb, Croatia; 5University of Zagreb, Faculty of Kinesiology, 10000 Zagreb, Croatia

**Keywords:** neuroendocrine neoplasms, gastrointestinal tract, pancreatic neuroendocrine neoplasms, grade, differentiation, liver metastases, liver transplantation, survival

## Abstract

Background/Objectives: Neuroendocrine neoplasms (NEN) are heterogeneous entities. Despite considerable advancement in the field, almost 50% of patients have metastatic disease, when liver transplantation (LT) is one of the possible treatments offering a cure in well-selected patients. Methods: The present study aims to systematically review all the literature from 2000 onwards on using LT for patients with NEN-LM, with particular attention to the risk factors of death and recurrence. Results: LT offers 5-year OS ranging from 52 to 74% and 5-year TFS rates ranging from 39 to 62%, with even better results published from 2009 onwards. The main risk factors for patient deaths are related to unfavorable primary tumor pathology, higher liver involvement, and simultaneous LT and primary resection. Similarly, recurrence is higher related to poor tumor grade and differentiation, and in the case of an older recipient age. Conclusions: Applying uniform criteria and a more in-depth understanding of the relevant prognostic factors contribute to a better selection of candidates for curative LT due to NEN metastases. LT for unresectable or liver-restricted NENs has a relevant place in the treatment algorithm and has achieved excellent results in recent decades, but more international efforts are needed to further improve outcomes.

## 1. Introduction

Neuroendocrine neoplasms (NENs) are an increasingly prevalent class of tumors most commonly arising in the gastroenteropancreatic tract, whose incidence shows an eight-fold increase over the last 50 years [1]. They are characterized by specific aspects like the presence of hormonally active (“functional”) tumors and propensity to metastasize despite their generally accepted relatively indolent growth [2]. Because of the latter, NENs are often diagnosed in advanced stages; this is particularly important as the median overall survival (OS) drastically drops from >30 years in localized stages to 10 years in regional disease to 12 months when distant metastases are present. Therefore, stage-dependent individualized therapy remains pivotal for long-term survival [3].

Liver transplantation (LT) for NEN liver metastases (NEN-LM) represents an efficacious treatment in well-selected patients in terms of OS and transplant benefit compared to more conventional therapies (i.e., hepatic resection) [4]. In 2007, the Milan criteria for NEN-LM were proposed by Mazzaferro et al. and then adopted by the USA United Network for Organ Sharing (UNOS) as well as by the European Neuroendocrine Tumor Society (ENETS). According to the Milan group, LT is considered a therapy in patients with NEN-LM with liver-only unresectable metastatic disease fulfilling the following specific inclusion criteria: (1) histological diagnosis of low-grade NET (i.e., low Ki-67 expression) regardless of the presence or absence of recognizable syndrome; (2) primary tumor located in the pancreas or intermediate gut (i.e., from distal stomach to sigmoid colon), thereby tributary of the portal vein, already removed with a curative resection; (3) < 50% of liver involvement; (4) stable disease for at least six months during the pre-LT period; (5) age < 55 years (relative contraindication) [5].

However, using LT to cure patients with NEN cannot be described as routine. For instance, only 188/127,851 (0.0015%) patients transplanted from 1988–2016 have been reported in the European Liver Transplant Registry [6].

Several potential risk factors for patient death and tumor recurrence after LT have been explored, investigating both tumor- and transplant-related features. Among them, we can report the histopathological assessment of proliferation markers (Ki-67 and/or mitotic indices) that are used for staging the tumor as low-grade (G1, Ki-67 ≤ 2%), intermediate-grade (G2, Ki-67 3–20%), or high-grade (G3, Ki-67 > 20%), the extent of liver disease, the location of the primary tumor, and the presence of a functional syndrome [7,8,9].

However, a detailed collection of the risk factors reported in the different articles has not been attempted yet. The present study aims to systematically review all the literature from 2000 onwards on using LT for patients with NEN-LM, with particular attention to the risk factors investigated in the different studies.

## 2. Materials and Methods

This systematic review was conducted and reported under the Preferred Reporting Items for Systematic Reviews and Meta-Analyses (PRISMA) and the guidance on conducting systematic reviews and meta-analyses of observational studies of etiology guidelines [10,11]. The study’s Prospero code is CRD42024587283.

### 2.1. Search Strategy and Eligibility Criteria for Studies

#### 2.1.1. Objective

The main goal of this review was to systematically analyze and summarize evidence relating to the role of liver transplantation for the cure of NEN and to identify the prognostic risk factors for patient death and tumor recurrence after transplantation.

#### 2.1.2. Search Strategy

The Medline (PubMed) database was searched for relevant published original articles using the following keywords: “liver OR hepat*” AND “transpl*” AND “neuroend*” through March 2024 by two authors (QL and AC), and in case of any potential conflict regarding the study selection consensus was made by a third reviewer (MCB), Figure 1. Included were studies in the English language, with no geographic or follow-up restrictions applied.

MEDLINE (PubMed) search strategy: (liver OR hepat*) AND (transpl*) AND (neuroend*)

Period of research: 1 January 2000–1 March 2024.

### 2.2. Eligibility Criteria

This review focused on retrospective and prospective observational studies that evaluated the post-LT outcomes of patients diagnosed with NEN-related liver metastases. All the studies exploring the impact of LT in the setting of NEN were included, with particular attention to the ones investigating the prognostic factors for post-LT death or tumor recurrence.

Case series with less than 10 cases reported, case reports, or literature reviews were not included. A limitation in the year of publication was applied, excluding all the studies before January 2000. The decision to use this cut-off was derived from the necessity of minimizing potential biases derived from including the very elderly series. With the intent to not exclude papers coming from the same centers or with potential periods or patients overlapping exploring different risk factors for patient death or tumor recurrence, all the papers presenting the inclusion criteria were reported no matter the potential overlapping.

### 2.3. Data Extraction

Information extracted in each selected study included first author (reference number), reference, year of publication, country, number of cases, period or study enrollment, male sex, age, site of the primary NEN, extension of the liver involvement caused by the NEN metastases, Ki-67 index, tumor grade 3, 5- and 10-year OS and tumor-free survival (TFS) rates, risk factor correlated with patient death or NEN recurrence, and statistical relevance extracted according to the statistical method used in the article (Cox regression, logistic regression, Kaplan–Maier analysis).

### 2.4. Risk of Bias Assessment

The risk of bias in the included studies was evaluated using the Risk Of Bias In Non-randomized Studies—f Exposure (ROBINS-E) tool [12], and the results of ROBINS-E assessment were visualized with the Robvis tool [13]. The validity of studies was judged for seven potential bias domains: (1) confounding, (2) measurement of the exposure, (3) selection of participants into the study (or into the analysis), (4) post-exposure interventions, (5) missing data, (6) measurement of the outcome, and (7) selection of the reported result. Each bias domain was assessed using a series of signaling questions. The risk of bias was rated as low, with some concerns, or high. For concerns not applicable to the studies, we assumed that there were no domain issues. After assessing all seven bias domains, an overall judgment was provided. Data are shown in Figure 2.

## 3. Results

The search strategy identified 793 records, with no records identified from the reference lists. Seven hundred and ten records were excluded because they were irrelevant to the review question or did not adhere to the inclusion criteria. Of the remaining 83 eligible records, 56 full-text articles were discarded for several reasons (Figure 1). In detail, the reasons for discarding were as follows: study not relevant (*n* = 2), study with insufficient data (*n* = 4), review article (*n* = 31), case report or case series with less than 10 cases (*n* = 17), letter or editorial (*n* = 2).

From the selection process, the pool of studies for qualitative synthesis was reduced to 27 [14,15,16,17,18,19,20,21,22,23,24,25,26,27,28,29,30,31,32,33,34,35,36,37,38,39,40]. Key characteristics of the studies included in the systematic review are illustrated in Table 1.

None of the included studies were prospective or randomized controlled trials, while all were retrospective experiences. Several studies reported national or international registry data [16,19,21,26,27,29,30,31,32]. Only one study reported balanced results after propensity score analysis [25].

Studies were conducted in 13 countries: the USA (*n* = 6), Sweden (*n* = 4), Germany (*n* = 2), Italy (*n* = 2), Poland (*n* = 2), Brazil (*n* = 1), Croatia (*n* = 1), Iran (*n* = 1), Belgium (*n* = 1), Hungary (*n* = 1), France (*n* = 1), Spain (*n* = 1), and the Netherlands (*n* = 1). One international study and two studies coming from European registries were also reported.

The study population ranged from 10 to 225 participants. Several overlapping data were reported. For example, the Goteborg experience was reported in three articles [33,37,39], in which the global number of reported cases was 15 patients. Similarly, the numerous US national experiences based on the UNOS database [21,26,27,30,31] reported similar experiences in which an evident overlapping of the patients was present. Also, the experiences from the Istituto Tumori Milan, Italy [18,25], and the Warsaw Center, Poland [14,28] showed similar overlapping. Finally, some European and international registries probably incorporate the cases reported in the monocentric series. Consequently, it is difficult to report the exact number of cases reported worldwide, but we can assume that this number does not surpass 500 cases.

The mean or median patient age range was 32.1–51.2 years, and the percentage of males ranged from 27.3 to 80.0%. The site of the primary tumor was reported in 22 articles. Also, these data were complex to report due to the significant overlapping. After having removed the international studies and having included only the most recent studies coming from the same center or national registry [14,15,17,18,20,22,23,24,26,32,33,34,35,36,40], the duodenum/pancreas resulted as the most common site of the primary tumor (*n* = 178/381, 46.7%), followed by the small bowel (*n* = 124/381, 32.5%), the sigmoid colon/rectum (*n* = 9/381, 2.4%), and the stomach (*n* = 7/381, 1.8%). Other/undetected/unknown sites were relatively common (63/381, 16.5%).

In the studies reporting the histological details of the tumors, the mean/median value of Ki-67 ranged from 0.7 to 15.8%. Five-year OS rates ranged from 36.0 to 97.2%, with a median of 64.7%. Ten-year survivals ranged from 46.1 to 93.0% (median = 70.4%). As for the TFS rates, the 5-year values ranged from 20.0 to 86.9% (median = 43.0%). When the 10-year survivals were reported, they ranged from 15.5 to 86.9% (median = 30.8%).

### 3.1. Risk Factors for Post-Transplant Death

A significant heterogeneity was reported in terms of the identification of the risk factors for post-LT death. Overall, it was possible to classify them into four groups, namely (a) the pathological aspects, (b) the extent of the metastatic disease, (c) the extent and management of the primary tumor, and (d) the transplant-related aspects (Table 2).

As for the pathological aspects, they principally focused on the mitotic activity, looking at the impact of Ki-67 and the tumor grading. Five studies explored these aspects [14,16,26,29,40]. As expected, both the Ki-67 and grading increase were correlated with a higher risk of death. In detail, considering the Ki-67 as a continuous variable, Kuncewicz identified an HR = 1.17 (*p* = 0.035) [14]. Considering it as a categorical variable, Eshmuminov reported an HR = 2.52 (*p* = 0.021) in the case of tumors with G2 vs. G1, namely corresponding to an increased risk in patients with a Ki-67 ≤ 2% vs. 3–20% [16]. Of note, this study was the only one with enough numerosity (*n* = 225) to consent to performing a multivariable Cox regression analysis. Similarly, Le Treut reported a significantly worse survival in patients with a G3 (i.e., Ki-67 > 20%), with a 5-year rate of 27.0% (*p* < 0.01) [29]. Sher et al., exploring an extensive US database, reported that the increase in tumor grading corresponded to an HR = 3.1 (*p* = 0.003) for the risk of death [26].

Only limited experiences reported the impact of lymph-nodal positivity, macrovascular invasion, and E-cadherin staining as risk factors for patient death [26,40].

As for the studies exploring the extent of metastatic disease at the level of the liver, Le Treut investigated this relevant aspect. In detail, the multicentric study from France published in 2008 generically explored the impact of the variable “hepatomegaly”, reporting a multivariable HR = 2.63 (*p* = 0.02). The large experience from the Liver Transplant European Registry confirmed the impact of this parameter, with a 5-year rate of 39.0% (*p* < 0.00001) in patients with hepatomegaly. Similarly, tumor bulk as an indication for LT and an estimated invasion of the liver overpassing 50% were relevant negative predictors [29]. The involvement > 50% was incorporated into the Milan criteria proposed by Mazzaferro. According to this aspect, the large study from Eshmuminov et al. explored the adverse impact of being transplanted out of the conventional Milan criteria, reporting a multivariable HR = 2.40 (*p* = 0.018) [16].

As for the extent and characteristics of the primary NEN, the necessity to perform a multivisceral transplantation corresponded to an increased risk for the outcome. In detail, Le Treut reported that the necessity to perform an upper abdominal exenteration correlated with an increased HR (multivariable: 3.72; *p* = 0.0034) for the risk of death [32]. Similarly, Sher reported in a US experience that performing an LT vs. an extensive need for resection for the transplant or removing the primary tumor was a significant protective factor (univariable HR = 0.81; *p* = 0.007).

Another relevant aspect concerning the primary site of the tumor was the possibility of removing it at a different time concerning the LT: Le Treut reported significantly inferior survivals in the cases in which the primary tumor was identified and removed contemporaneously with the transplant (5-year rates: 22.0 vs. 56.0%; *p* < 0.00001), or in the cases in which a major resection in the site of the primary tumor was needed in addition to LT (5-year rates: 21.0%; *p* < 0.000001) [29]. Lastly, the site of the primary tumor was also explored, reporting that the tumor involving the duodenum/pancreas was more commonly correlated with increased survival (multivariable relative risk = 2.94; *p* = 0.0018), probably due to the increased surgical complexity observed in the case of duodenopancreasectomy [32].

Some experiences, mainly from the US, explored the risk factors correlated with the transplant [21,27,30]. Valvi et al. identified some risk factors for patient death directly connected with the transplant procedure: recipient age (multivariable HR = 1.013; *p* < 0.001), donor age (multivariable HR = 1.007; *p* < 0.001), cold ischemia time (multivariable HR = 1.013; *p* = 0.003), and MELD (multivariable HR = 1.016; *p* < 0.001) [21]. As expected, tumor recurrence also correlated with an increased risk of death (multivariable HR = 4.256; *p* < 0.001) [21].

Nobel explored the relevance of the recipient total bilirubin value, observing that patients arriving at LT with a value > 1.3 mg/dL showed markedly inferior survivals (3-year rates: 78.3 vs. 36.4%; *p* = 0.005 [27]. Nguyen et al. also observed that the total bilirubin of the recipient correlated with poor outcomes, with an OR = 1.063 (*p* = 0.02). Similarly, the same authors observed that recipient albumin (OR = 0.480; *p* = 0.011) and donor creatinine (OR = 1.288; *p* = 0.004) were also risk factors for patient death [30].

Some papers compared the outcomes of the NEN compared to other LT oncological indications. In detail, two studies compared the results of post-LT OS in NEN vs. HCC cases. Interestingly, no statistical relevance was reported, showing the good survival results observed in well-selected NEN patients [19,31].

Lastly, two studies explored the role of LT vs. hepatic resection. Mazzaferro et al., using a propensity score-adjusted model, reported that resected patients had an HR = 7.4 (*p* = 0.001) compared to the transplanted patients, showing the benefit of being transplanted in the case of correct selection [25]. The same group similarly explored the same concept, showing the protective effect of LT vs. resection (HR = 0.35; *p* = 0.01) [18].

### 3.2. Risk Factors for Post-Transplant Recurrence

In addition, significant heterogeneity was also reported when the risk factors for post-LT recurrence were identified. Also in this case, it was possible to classify some groups: (a) the pathological aspects and (b) the transplant-related aspects (Table 3).

As for the pathological aspects, the mitotic activity was correlated with worse results in the recurrence setting [14,28]. In detail, considering the Ki-67 as a continuous variable, Kuncewicz et al. identified it as a risk factor for recurrence, with an univariable HR = 1.20 (*p* = 0.022) [14]. Grąt et al. similarly observed that a Ki-67 > 2% correlated with a 5-year TFS = 0.0% (*p* = 0.048) [28].

As for the transplant-related factors, recipient age was a relevant factor for the risk of recurrence: Kuncewicz identified an univariable HR = 5.47 (*p* = 0.046) for the risk of recurrence in the case of a recipient age ≥55 years [14]. Valvi also identified the recipient’s age as a risk factor (multivariable HR = 1.008; *p* < 0.001) [21]. Additionally, Valvi also identified other risk factors for recurrence: donor age (multivariable HR = 0.991; *p* = 0.001), recipient BMI (multivariable HR = 1.007; P < 0.001), cold ischemia time (multivariable HR = 1.017; *p* < 0.001), and MELD (multivariable HR = 1.011; *p* < 0.001) [21]. Interestingly, Grąt identified the need for intraoperative blood transfusions as a risk factor for recurrence, with 5-year TFS rates of 30.0 vs. 100.0% (*p* = 0.021) [28].

Valvi explored the risk for recurrence of NEN compared with other tumors (i.e., CCA and HCC), reporting a reduced risk in the case of NEN (multivariable HR = 0.412 (*p* < 0.001) [21]. Lastly, Maspero compared LT vs. resection, reporting that, in well-selected NEN cases, the risk of recurrence after LT was inferior to the resection (multivariable HR = 0.36; *p* < 0.0001 [18].

## 4. Discussion

Understanding how to manage NENs is essential for optimizing the treatment. Patients’ outcomes depend on tumor location, size, and distribution, as well as on the presence and extent of hepatic metastases and tumor growth rate. Unfortunately, by the time of diagnosis, over 50% of patients already have metastatic disease. Therefore, although far from routine, LT plays a relevant role in the treatment algorithm thanks to its potential to offer a cure even in metastatic patients [41].

According to the results observed in our systematic review, it represents an excellent strategy in well-selected patients as the cure for NEN liver metastases. In detail, 5-year OS ranged 52–74% in large multicentric experiences published in the last ten years [16,26], with some monocentric series showing even 97% of survival rates [25]. Similarly, 5-year TFS rates ranged from 39 to 62% [16,29], with, also in this case, some excellent monocentric results (i.e., 87% in the Milan series) [25]. After looking at the reports of different series, some impaired results were observed, mainly in the case of recurrence rates. There are few possible explanations to justify these results. A study exploring the survivals in patients with NEN receiving different treatments showed that those diagnosed more recently (i.e., between 2009 and 2012) had better OS when compared to the patients diagnosed between 2000 and 2004 (HR = 0.79, 95%CI = 0.73–0.85), which, in addition to advancements in NEN therapeutics related to the introduction of somatostatin receptor antagonists, everolimus, and sunitinib, might be influenced by the success of LT [42]. Similarly, the poor results sometimes observed in the LT series are caused by the presence of historical cases (i.e., LT before 2000) included in the studies with the intent to increase their sample size [29,31,33]. The European series published by Le Treut et al. specifically analyzed this aspect, reporting a relevant increase in the 5-year OS and TFS rates comparing the patients transplanted during the period 1982–1999 vs. 2000–2009 (46.0 vs. 59.0% and 30.0 vs. 39.0%) [29]. Therefore, we can assume that more recently transplanted cases effectively reported better survivals. Transcribing this concept into survival gain, Mazzaferro et al. reported that patients with LT had a marked benefit compared to non-transplanted ones, with gains of 6.8 months at five years and 38.4 months at ten years [25].

In light of this evidence, it is clear that the opportunity to guarantee well-selected cases the possibility to be transplanted represents an extraordinary opportunity for survival. Therefore, identifying the risk factors for patient death or recurrence is of capital relevance. In this study, several potential contributing factors reducing death and NEN recurrence after LT have been identified.

Unfortunately, the international literature is sparse in terms of investigated variables. Only a few studies focused on the same variables, and many of them were only based on small monocentric series. A small percentage of studies (*n* = 3/27, 11.1%) were based on international series [16,19,29], with the largest cohort only based on 225 cases. The mentioned data stress the main problem in the setting of LT and NEN, namely the absence of a real international effort to create a large database able to investigate the different features with complex statistical approaches.

Overall, the most explored features in the setting of transplantation were related to the pathology (i.e., Ki-67 and grading), the extent of the metastatic disease, the management of the primary NEN, and the LT-related aspects (i.e., recipient age) [14,16,21,26,28,29,30,32,40].

According to the Milan criteria, only patients with Ki-67 ≤ 2% (namely G1) can be considered for LT. As already reported, only one study, thanks to its numerosity (*n* = 225), was able to perform a multivariable analysis specifically investigating the risk for death in patients with G2 vs. G1 (HR = 2.52, P = 0.021) [16]. The arbitrary cut-off of 3% for Ki-67 represents a definitive threshold value for enrolling a patient for LT. On the other hand, no study had enough statistical power to explore this aspect in detail. A large systematic review performed in patients with non-metastatic NEN receiving surgery reported several thresholds ranging from 2 to 20% [43]. A study from the Netherlands (*n* = 241) reported a cut-off of 5%, with excellent OS and TFS rates despite this slight enlargement in Ki-67 [44]. Similarly, a study from Italy (*n* = 274) suggested that improvement of the prognostic value of NEN grading could be achieved with an increase of Ki-67 from the traditional cut-off value for the G2 stage of 2% to 5% [45]. These analyses have been performed in no-metastatic cases, and large international studies are necessary to explore if a similar increase in the Ki-67 should be considered in the setting of LT.

In the present day, the dimension of the metastatic burden must be considered in light of a modern decision-making approach in which newly available agents beneficially influence tumor cell growth and contribute to the reduction of metastatic load [46]. Available data suggest that < 50% of liver tumor invasion offers better survival compared to >50% of hepatic involvement [29]. However, such a cut-off represents a gross threshold for selecting patients, mainly because the use of modern strategies might be beneficial in reducing the hepatic tumor load, thus improving the success of the LT. Therefore, the radiological response after treatment [47] or the ability to downstage the tumor to transplantability criteria, like in the setting of hepatocellular carcinoma, should be considered and investigated in more detail [48].

Therefore, it is relevant that oncologists always consider the LT opportunity for patients with metastatic NEN, discussing these patients in multidisciplinary meetings with a surgeon and a hepatologist expert in LT. LT timing remains an important issue. ELTR data analysis revealed that the time between diagnosing NEN metastases and the number of patients undergoing surgical procedures and locoregional therapies before LT increased after 2000 [29]. Therefore, new therapies must dictate the treatment sequencing strategies without interfering with or even precluding the LT opportunity for these patients [49].

Reports suggest that an LT performed after removing the primary tumor presents better survival [26,29]. Resection of the primary tumor before LT allows a better understanding of tumor biology and reduces the residual tumor aggressiveness after primary resection [50]. On the other hand, LT should be avoided simultaneously with major extrahepatic resection [29]. This aspect might be especially important in patients with pancreatic NENs, in which worse OS and higher tumor recurrence rates have been observed, mainly when associated with high 5-hydroxyindoleacetic acid (5-HIAA )levels and a Ki-67 ≥ 5% [35].

Lastly, several LT-related features have been explored in the reported literature, but only the recipient age in more detail. The cut-off initially proposed by Mazzaferro et al. of >55 years [25] should nowadays be considered obsolete, with several studies exploring the possibility of transplanting very elderly patients [51,52]. No studies exist explicitly looking at the impact of LT on elderly recipients transplanted in recent years. Therefore, we can assume that the routinely adopted age cut-off (i.e., 70 years) could also be considered in the setting of NEN metastases.

Considering the results observed in the present systematic review, we can assume that the biggest challenge to overcome as a future step is the creation of projects investigating proper clinical data on larger populations. Approximately 20 LT/year reported in the last 25 years only represents a part of the cases transplanted worldwide. The transplant community should make a great effort with the intent to create international databases able to improve our knowledge of this pathology.

LT for NEN metastases remains a debated topic, representing a niche treatment today. Nowadays, several strategies beyond LT can be considered. These include surgical resection for patients with resectable metastases, which can offer long-term survival benefits. Locoregional therapies like transarterial chemoembolization, radiofrequency ablation, or selective internal radiation therapy may be used to control tumor growth. Systemic therapies such as somatostatin analogs, peptide receptor radionuclide therapy, and targeted therapies like everolimus or sunitinib can help manage unresectable or progressive disease. Additionally, liver-directed therapies, including hepatic artery embolization, can palliate symptoms and slow tumor progression.

However, in the next 10–20 years, the role of LT may expand, particularly in centers with expertise in both transplant surgery and neuroendocrine oncology. Specifically, three areas of evolution should be considered for future perspectives on LT for the treatment of NEN metastases. First, the refinement of selection criteria, with the inclusion of more precise molecular and genetic markers able to help improve patient selection and outcomes. Second, more detailed research on tumor biology is able to give a deeper understanding of NEN biology and the interaction of its microenvironment with the immune system. Third, the role of the multidisciplinary approaches, integrating LT into a multimodal treatment strategy combining surgery, loco-regional therapies, and targeted therapies.

The current analysis presents some limitations. One of them is that LT has not been compared in terms of results with other efficacious therapies, apart from a study in which transplantation has been compared with resection [18]. No studies specifically explored the efficacy of LT vs. somatostatin. Studies focused on this topic or a network meta-analysis are required in this setting.

Another limitation of the current study is the absence of more in-depth studies exploring specific parameters like age, diet, gender, or geographical location. We can presume that such parameters can relevantly impact tumor aggressiveness and recurrence risk. Consequently, studies exploring more data might be required in the future to come up with a holistic view of this phenomenon.

## 5. Conclusions

Applying uniform criteria (such as Milan) and a more in-depth understanding of the relevant prognostic factors contribute to a better selection of patients who are candidates for curative liver transplantation due to NEN metastases. Liver transplantation for unresectable or liver-restricted NENs has a relevant place in the treatment algorithm and has achieved excellent results in recent decades. Some parameters adopted for selecting patients with LT should be cautiously expanded or rediscussed. Therefore, an international effort is required to create a large database to improve the knowledge in this setting.

## Figures and Tables

**Figure 1 biomedicines-12-02419-f001:**
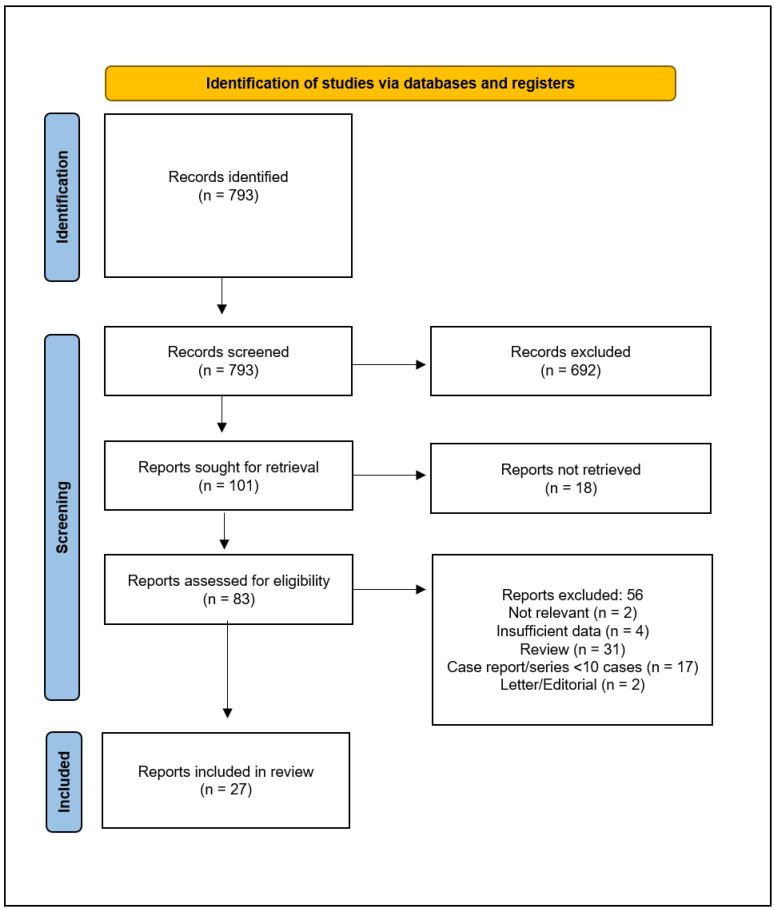
Flow diagram for study selection.

**Figure 2 biomedicines-12-02419-f002:**
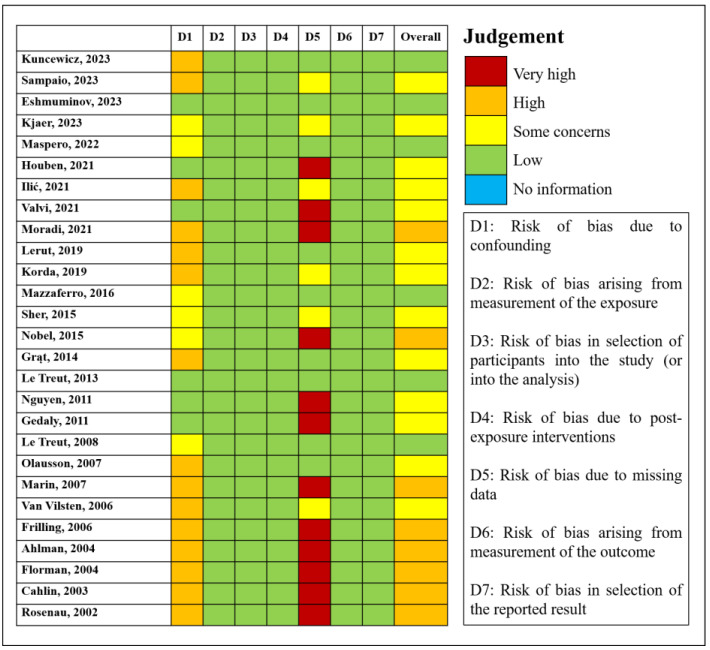
Presentation of risk of bias for the included studies.

**Table 1 biomedicines-12-02419-t001:** Summary of included studies regarding liver transplantation for the management of neuroendocrine neoplasms.

Name (Ref), Year	Country	N	Period	Male	Mean Age, Years	Site Primary	Mean Extent Liver Involvement (%)	Mean Ki67 Index (%)	Grade 3	OS		TFS	
										5-Year	10-Year	5-Year	10-Year
Kuncewicz [14] 2023	Poland	19	1989–2022	11	51 *	Pancreas 8 Small bowel 6 Colon 3 Stomach 1 Unknown 1	7.5 *	2 *	-	88.0	70.4	64.8	27.8
Sampaio [15] 2023	Brazil	11	2001–2021	3	45	Gastrointestinal 7 Cervix 1 Unknown 3	-	Ranges 1–5	-	45.4	-	36.3	-
Eshmuminov [16] 2023	Europe	225	-	120	47 *	Small bowel 101 Pancreas 80 Other 44	17.3 *	3	10	74.0	-	62.4	-
Kjaer [17] 2023	Sweden	41	2000–2021	20	-	Pancreas 41	-	-	0	64.7	-	-	-
Maspero [18] 2022	Italy	48	1991–2019	30	44	Small bowel 31 Pancreas 14 Duodenum 2 Stomach 1	<25% = 20 25–50% = 21 >50% = 7	3	2	95.5	93.0	75.0	52.0
Houben [19] 2021	International	100	1988–2017	47	45.4	-	-	-	-	55.7	-	-	-
Ilić [20] 2021	Croatia	12	2009–2020	6	47.8	Pancreas 7 Small bowel 2 Other 3	-	9.6	2	91.7	-	66.7	-
Valvi [21] 2021	USA	206	1988–2018	117	48.2	-	-	-	-	64.9	46.1	43.9	-
Moradi [22] 2021	Iran	15	2011–2017	6	32.1	Pancreas 5 Small bowel 4 Unknown 6	-	-	-	53.3	-	-	-
Lerut [23] 2019	Belgium	18	1987–2016	-	47.8	Small bowel 5 Pancreas 11 Stomach 1 Other 1	<25% = 8 25–50% = 4 >50% = 6	15.8	-	75.8	56.8	33.8	33.8
Korda [24] 2019	Hungary	10	1995–2018	4	50.0	Pancreas 3 Small bowel 3 Other 2 Unknown 2	-	Ranges 1–25	0	71.0	-	43.0	-
Mazzaferro [25] 2016	Italy	42	1995–2010	26	40.5 *	Ileum/cecum 24 Duodenum/pancreas 15 Colon/stomach 3	<25% = 17 25–50% = 21 >50% = 4	≤3% = 33 >3% ≤ 20% = 7	1	97.2	88.8	86.9	86.9
Sher [26] 2015	USA	85	1988–2012	51	48.0 *	Duodenum/Pancreas 42 Digestive Tract 24 Other/Unknown 19	-	-	5	52.0	-	-	-
Nobel [27] 2015	USA	108	2002–2014	-	-	-	-	-	-	63.0	-	-	-
Grąt [28] 2014	Poland	12	1989–2013	4	47.0 *	Pancreas 6 Small bowel 3 Colon 2 Unknown 1	7.0%	2 *	0	78.6	78.6	51.6%	15.5% (9 yrs)
Le Treut [29] 2013	Europe	213	1982–2009	114	46.0	Bronchial tree 16 Stomach 8 Duodenum 3 Pancreas 94 Jejunum 16 Ileum 48 Right colon 4 Sigmoid colon 1 Rectum 5 Common bile duct 1 Undetected 17	<20% = 31 20–40% = 41 40–60% = 41 60–80% = 54 >80% = 31 Unknown = 13 50% *	-	-	52.0	-	30.0	-
		106	2000–2009	-	-	-	-	-	-	59.0	-	39.0	-
Nguyen [30] 2011	USA	184	1988–2011	-	-	-	-	-	-	49.2	-	-	-
		110	2002–2011	-	-	-	-	-	-	58.0	-	-	-
Gedaly [31] 2011	USA	150	1988–2008	84	45.1	Carcinoid 51 Insulinoma 6 Glucagonoma 3 Gastrinoma 11 VIPoma 9 unspecified NET 70	-	-	-	49.0	-	32.0	-
Le Treut [32] 2008	France	85	1989–2005	46	45.0	Bronchial tree 5 Stomach 3 Jejunum 6 Ileum 16 Sigmoid colon/rectum 4 Duodenum/pancreas 41 Undetected 10	hepatomegaly in 53/85 cases 50% *	-	-	47.0	-	20.0	-
Olausson [33] 2007	Sweden	15	1997–2005	11	49.9	Pancreas 10 Midgut carcinoid 2 Hindgut cardinoid 1 Foregut cardinoid 1 Unknown 1	In all the cases > 50%	3.7	0	90.0 LT only; 71.0 entire cohort	-	20.0	-
Marin [34] 2007	Spain	10	1996–2006	5	42.0	Pancreas 6 Small bowel 2 Lung 2	-	-	-	57.0 (3-yr)	-	33.0 (recurr rate)	-
Van Vilsten [35] 2006	Netherlands	19	1998–2004	15	47.0 *	Pancreas 11 Ileum 5 Small bowel 2 Liver 1	-	0.69 <2% = 15 >2% = 3 Unknown = 1	-	87.0 (1-yr)	-	77.0 (1-yr)	-
Frilling [36] 2006	Germany	15	1992–2004	10	49.8 *	Pancreas 7 Ileum 3 Lung 1 Colon 1 Bronchus 2 Unknown 1	-	-	-	67.2	-	48.3	-
Ahlman [37] 2004	Sweden	12	1997–2004	9	50.0 *	Pancreas 8 Midgut carcinoid 2 Hindgut cardinoid 1 Foregut cardinoid 1	-	5.0	0	2 deaths	-	5 recurr	-
Florman [38] 2004	USA	11	1992–2002	4	51.2	Pancreas 7 Appendix 1 Ileum 1 Rectum 1 Unknown 1	-	-	-	36.0	-	-	-
Cahlin [39] 2003	Sweden	10	1997–2003	8	47.0 *	Pancreas 6 Ileum 2 Rectum 1 Lung 1	-	6.2	0	2 deaths	-	5 recurr	-
Rosenau [40] 2002	Germany	19	1982–1997	9	44.0 *	pancreas 10 small bowel 6 stomach 1 lung 1 unknown 1	-	4.8	-	80.0	50.0	21.0	21.0

* Median. Abbreviations: Ref, reference; N, number; OS, overall survival, TFS, tumor-free survival; LT, liver transplantation.

**Table 2 biomedicines-12-02419-t002:** Risk factors for patient death reported in the included studies.

Author (Ref)	Risk Factors	Statistical Relevance
**Pathological aspects**		
Kuncewicz [14]	Ki-67	Univariable HR = 1.17 (95%CI = 1.01–1.35), *p* = 0.035
Rosenau [40]	Ki-67	Univariable Cox regression; *p* = 0.01
Eshmuminov [16]	Tumor grade G2	Multivariable HR = 2.52 (95%CI = 1.15–5.52), *p* = 0.021
Sher [26]	Tumor grading	Univariable HR = 3.1 (95%CI = 1.5–6.6), *p* = 0.003
Le Treut [29]	Poorly differentiated tumor	5-year rates: 27.0%; *p* < 0.01
Rosenau [40]	Aberrant staining E-cadherin	Univariable Cox regression; *p* = 0.01
Sher [26]	Large vessel invasion	Univariable HR = 12.0 (95%CI = 3.7–42.0), *p* < 0.001
Rosenau [40]	Lymph-nodal positivity	Univariable Cox regression; *p* = 0.02
**Extent of the metastatic disease at the level of the liver**		
Le Treut [29]	Hepatomegaly	5-year rates: 39.0%; *p* < 0.00001
Le Treut [32]	Hepatomegaly	Multivariable RR = 2.63 (95%CI = 1.20–5.80, *p* = 0.0157
Le Treut [29]	Tumor bulk as indication for LT	5-year rates: 35.0%; *p* < 0.0001
Le Treut [29]	Estimated tumoral invasion > 50% vs. < 50%	5-year rates: 42.0 vs. 61.0%; *p* < 0.002
Eshmuminov [16]	Outside MC	Multivariable HR = 2.40 (95%CI = 1.16–4.92), *p* = 0.018
**Extent and management of the primary tumor**		
Sher [26]	Extent of primary resection	Univariable HR = 0.81 (95%CI = 0.70–0.94), *p* = 0.007
Le Treut [29]	Resection of primitive during LT (vs. other times)	5-year rates: 22.0 vs. 56.0%; *p* < 0.00001
Le Treut [29]	Major resection in addition to LT	5-year rates: 21.0%; *p* < 0.000001
Le Treut [32]	Upper abdominal exenteration	Multivariable RR = 3.72 (95%CI = 1.54–8.95), *p* = 0.0034
Le Treut [32]	Primary tumor site in duodenum/pancreas	Multivariable RR = 2.94 (95%CI = 1.49–5.79, *p* = 0.0018
**Transplant-related variables**		
Valvi [21]	Recipient age	Multivariable HR = 1.013 (95%CI = 1.009–1.016), *p* < 0.001
Valvi [21]	Donor age	Multivariable HR = 1.007 (95%CI = 1.005–1.009), *p* < 0.001
Valvi [21]	CIT	Multivariable HR = 1.013 (95%CI = 1.005–1.022), *p* = 0.003
Valvi [21]	MELD	Multivariable HR = 1.016 (95%CI = 1.012–1.019), *p* < 0.001
Valvi [21]	Recurrence	Multivariable HR = 4.256 (95%CI = 3.988–4.542), *p* < 0.001
Nobel [27]	Recipient bilirubin≤ vs. >1.3 mg/dL	3-year rates: 78.3 vs. 36.4%; *p* = 0.005
Le Treut [29]	Period 1982–1999 vs. 2000–2009	5-year rates: 46.0 vs. 59.0%; *p* < 0.05
Nguyen [30] §	Recipient total bilirubin	OR = 1.063, *p* = 0.02
Nguyen [30] §	Recipient albumin	OR = 0.480, *p* = 0.011
Nguyen [30] §	Donor creatinine	OR = 1.288, *p* = 0.004
**NET vs. other tumors**		
Gedaly [31]	NET vs. HCC	5-year rates: 48.0 vs. 58.0%; *p* = 0.23
Houben [19]	NET vs. HCC	Univariable HR = 1.27 (95%CI = 0.92–1.77), *p* = 0.15
**Transplant vs. resection**		
Maspero [18]	LT vs. resection	Multivariable HR = 0.35 (95%CI = 0.16–0.78), *p* = 0.010
Mazzaferro [25]	LT vs. resection	Multivariable HR = 7.4 (95%CI = 2.4–23.0), *p* = 0.001 (PS-adjusted)

§ LT after 2002. **Abbreviations:** Ref, reference; HR, hazard ratio, CI, confidence intervals; RR, relative risk; LT, liver transplantation; MC, Milan criteria; CIT, cold ischemia time; MELD, model for end-stage liver disease; NET, neuroendocrine tumor; HCC, hepatocellular carcinoma; OR, odds ratio; PS, propensity score.

**Table 3 biomedicines-12-02419-t003:** Risk factors for tumor recurrence reported in the included studies.

Name (Ref)	Risk Factors	Statistical Relevance
**Pathological aspects**		
Kuncewicz [14]	KI-67	Univariable HR = 1.20 (95%CI = 1.03–1.41), *p* = 0.022
Grąt [28]	Grading (G2 vs. G1)	5-year rates: 20.8 vs. 80.0%; *p* = 0.037
Grąt [28]	KI-67 (> vs. <2%)	5-year rates: 0.0 vs. 83.3%; *p* = 0.048
**Transplant-related variables**		
Kuncewicz [14]	Age ≥ 55 years	Univariable HR = 5.47 (95%CI = 1.03–29.08), *p* = 0.046
Valvi [21]	Recipient age	Multivariable HR = 1.008 (95%CI = 1.005–1.012), *p* < 0.001
	Donor age	Multivariable HR = 0.991 (95%CI = 0.985–0.996), *p* = 0.001
	Recipient BMI	Multivariable HR = 1.007 (95%CI = 1.006–1.009), *p* < 0.001
	CIT	Multivariable HR = 1.017 (95%CI = 1.008–1.025), *p* < 0.001
	MELD	Multivariable HR = 1.011 (95%CI = 1.008–1.015), *p* < 0.001
Grąt [28]	Intraoperative blood transfusions (yes vs. no)	5-year rates: 30.0 vs. 100.0%; *p* = 0.021
**NET vs. other tumors**		
Valvi [21]	NET vs. other tumors (CCA and HCC)	Multivariable HR = 0.412 (95%CI = 0.331–0.512), *p* < 0.001
**Transplant vs. resection**		
Maspero [18]	LT vs. resection	Multivariable HR = 0.36 (95%CI = 0.21–0.59), *p* < 0.0001

**Abbreviations:** Ref, reference; HR, hazard ratio, CI, confidence intervals; BMI, body mass index; CIT, cold ischemia time; MELD, model for end-stage liver disease; NET, neuroendocrine tumor; CCA, cholangiocellular carcinoma; HCC, hepatocellular carcinoma.

## Data Availability

No new data were created.

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
