# Peer review of "Liver Transplantation for the Cure of Neuroendocrine Liver Metastasis: A Systematic Review with Particular Attention to the Risk Factors of Death and Recurrence"

_biomedicines, 2024, doi:10.3390/biomedicines12112419_

Round 1
Reviewer 1 Report
Comments and Suggestions for Authors
The authors present a systematic review of collection of literatures involving liver cancer and liver transplant. The authors have properly catagorized different parameters from existing literatures and showed a correlative trend between liver metastatsis and liver transplantation.
I would suggest the authors to have bit more discussion on this topic, related to future steps. The biggest challenge for such projects is non-availability of proper clinical data. Even though in this manuscript the authors present 793 papers, the real numbers could be orders of magnitudes high in real world.
Many more parameters can affect such relations like age, diet, gender, geographical location, etc. Thus more parameters and importantly more data might be required in future to come up with the holistic view of this phenomenon.
Comments on the Quality of English LanguageOverall English is fine. The authors can check the grammar through the tools like Grammarly for more professional corrections.
Author Response
The authors present a systematic review of a collection of literature involving liver cancer and liver transplant. The authors have properly catagorized different parameters from existing literatures and showed a correlative trend between liver metastatsis and liver transplantation.
I would suggest the authors to have bit more discussion on this topic, related to future steps. The biggest challenge for such projects is non-availability of proper clinical data. Even though in this manuscript the authors present 793 papers, the real numbers could be orders of magnitudes high in real world.
Comments:
We completely agree with the Reviewer. We have added these points to the Discussion.
In light of the results observed in the present systematic review, we can assume that the biggest challenge to overpass as a future step is the creation of projects investigating proper clinical data on larger populations. Approximately 20 LT/year reported in the last 25 years only represents a part of the cases transplanted worldwide. The transplant community should make a great effort with the intent to create international databases able to improve our knowledge of this pathology.
Another limitation of the current study is the absence of more in-depth studies exploring specific parameters like age, diet, gender, or geographical location. We can presume that such parameters can relevantly impact tumor aggressiveness and the risk of recurrence. Consequently, studies exploring more data might be required in the future to come up with a holistic view of this phenomenon.
Reviewer 2 Report
Comments and Suggestions for Authors
Title: Liver transplantation for the cure of neuroendocrine liver metastasis: A systematic review with particular attention to the risk factors of death and recurrence
Thanks to the authors for this interesting review.
My comments:
1. In my opinion will it be possible authors add some few sentences in their discussion about other strategies that could help with liver metastasis aside LT?
2. Could authors also provide some future perspectives on LT. Where do authors see this approach in the next 10-20 years?
Author Response
Title: Liver transplantation for the cure of neuroendocrine liver metastasis: A systematic review with particular attention to the risk factors of death and recurrence
Thanks to the authors for this interesting review.
My comments:
- In my opinion will it be possible authors add some few sentences in their discussion about other strategies that could help with liver metastasis aside LT?
RESPONSE:
We thank the Reviewer for the comment. We have added these sentences in the Discussion.
For NEN liver metastases, several strategies beyond LT can be considered. These include surgical resection for patients with resectable metastases, which can offer long-term survival benefits. Locoregional therapies like transarterial chemoembolization, radiofrequency ablation, or selective internal radiation therapy may be used to control tumor growth. Systemic therapies such as somatostatin analogs, peptide receptor radionuclide therapy, and targeted therapies like everolimus or sunitinib can help manage unresectable or progressive disease. Additionally, liver-directed therapies, including hepatic artery embolization, can palliate symptoms and slow tumor progression.
COMMENT 2:
Could authors also provide some future perspectives on LT. Where do authors see this approach in the next 10-20 years?
Response: We have added these considerations in the Discussion.
LT for NEN metastases remains a debated topic, representing a niche treatment today. However, in the next 10-20 years its role will expand, particularly in centers with expertise in both transplant surgery and neuroendocrine oncology. Over the next decades, three areas of evolution should be considered for the future perspectives on LT for the treatment of NEN metastases. First, the refinement of selection criteria, with the inclusion of more precise molecular and genetic markers able to help improve patient selection and positively impact the results. Second, more detailed research enabling a deeper understanding of NEN biology and the interaction of its microenvironment with the immune system is needed. Finally, the role of the multidisciplinary approaches, integrating LT into a multimodal treatment strategy combining surgery, loco-regional therapies, and targeted therapies should be considered.
